# Neural Relational Inference with Node-Specific Information

**Ershad Banijamali**
Amazon Alexa AI*
Toronto, Canada
ebanijam@amazon.com

## Abstract

Inferring interactions among entities is an important problem in studying dynamical systems, which greatly impacts the performance of downstream tasks, such as prediction. In this paper, we tackle the relational inference problem in a setting where each entity can potentially have a set of individualized information that other entities cannot have access to. Specifically, we represent the system using a graph in which the individualized information become node-specific information (NSI). We build our model in the framework of Neural Relation Inference (NRI), where the interaction among entities are uncovered using variational inference. We adopt NRI model to incorporate the individualized information by introducing *private nodes* in the graph that represent NSI. Such representation enables us to uncover more accurate relations among the agents and therefore leads to better performance on the downstream tasks. Our experiment results over real-world datasets validate the merit of our proposed algorithm.

## 1 Introduction

Our world includes many different types of systems that involve multiple entities interacting with each other, from biology to sports, from social media to driving situations. Modelling the behaviour of such dynamical systems is a challenging task, which requires uncovering different types of interactions among the entities and how they affect each other.

Recent approaches in machine learning learn the interactions among the entities through graph-based models and attention-based models, in which the representation of an entity is updated through its relationship with other entities. Such relationships are usually either predefined or uncovered by learning. Kipf et al. (2018) proposed neural relational inference (NRI) for relationship uncovering in the framework of variational inference (Kingma & Welling, 2013; Rezende et al., 2014). NRI has an encoder-decoder structure in which the latent codes represent different types of interactions among the entities. The distributions over the latent variables are inferred based on the input features of the entities and a graph structure is formed by sampling from these distributions. The decoder of the model is a GNN-based algorithm that runs over the uncovered graph to update the features of entities. The model shows prominent performance over many synthetic and real-world datasets.

Uncovering the interaction among entities in NRI and other models in the GNN literature is studied in problems where the features of the entities are completely observable and the GNN-based algorithm is also run on the observable features. However, in many real-world problems there is a set of hidden features that affect the way entities interact with each other. For example consider the problem of predicting the future position of vehicles in a driving scenario. In order to uncover the relation of a target vehicle with other vehicles not only should we consider the features extracted from the observations from vehicles, e.g. their trajectories up to the current time, but also we should take into account the intention of the target vehicle. In fact, intention, which can be an immediate action or a longer term goal, forms a set of features that is only accessible by the target vehicle and other vehicle cannot know it. In this paper we call such feature the individualized features or Node-Specific Information (NSI) in a graph representation of the system. Formally, NSI is a set of features that is

---

*Work was done prior to joining Amazon.

only accessible by one node in a graph structure but affects the interactions of that node with other nodes. Our goal is to efficiently exploit NSI to build more accurate graph structures and consequently achieve better performance in the downstream tasks. Towards this goal, the first step is to find a proper representation for NSI in our graph structure. We propose introducing a new set of nodes in the graph that carry NSI and call them *private nodes*. Observable features of the entities are then denoted by *public nodes*. Therefore each entity can be represented by a public and private node. We introduce our model in the framework of NRI, i.e. a variational inference model that uncovers different types of interaction among the entities. We carefully design the encoder and decoder part of our model to ensure that NSI remains an individual feature for one entity and does not affect the interaction modelling of the other entities. At the same time, through experiment we show that such modelling of NSI is very efficient and can result in significant improvement on the performance on downstream tasks. The main contributions of this work are the followings:

- To the best of our knowledge the problem of having individualized features for each entity has not been previously studied in the framework of relational inference. We tackle this problem by introducing a new set of nodes in a graph structure that represents the individualized information. These nodes are used during the process of relational inference as well as performing the downstream task, i.e. trajectory prediction in our case.
- We show that our proposed model can exploit NSI efficiently by introducing minimum additional computational complexity compared to the original NRI model and its variants.
- The results of our experiments on real-world datasets show that our model can outperform the baselines and achieve the state-of-the-art results on the defined tasks.

## 2 RELATED WORK

**Interaction modelling:** Relational learning is a popular approach for the problems with dependency structure among the data points. Such dependency can be predefined in some domains. But in many domains it has to be learned. For example approaches like locally linear embedding (LLE) (Roweis & Saul, 2000) and Isomap (Tenenbaum et al., 2000) use $k$NN for forming such relationships based on different measures of similarities among the data points. More recently, neural networks have become the dominant tools for learning these dependencies based on different architectures and paradigms (Kipf & Welling, 2017; Hamilton et al., 2017; Garcia Duran & Niepert, 2017; Monti et al., 2017; Veličković et al., 2017; Franceschi et al., 2019).

The most relevant works to our proposed model are NRI (Kipf et al., 2018) and dynamic NRI (dNRI) (Graber & Schwing, 2020), which are discussed in more details in the next section. Recently, Li et al. (2020) introduced similar ideas for multi-modal trajectories prediction. Extensions of NRI in other directions than ours also appeared in the literature. For example, Li et al. (2019) tries to uncover interactions by imposing some structural constraints on the prior and Webb et al. (2019) introduces the idea of factorized graph for NRI.

The other common approach for uncovering relationships among the entities is based on the idea of attention. This idea has been used in Narasimhan et al. (2018); Hoshen (2017); Van Steenkiste et al. (2017); Garcia & Bruna (2017); Monti et al. (2017); Veličković et al. (2017), where the attention mechanism is the main tool for interaction uncovering, however, it is also used as a building block for GNNs.

**Future trajectory prediction as evaluation metric:** We define our problem as uncovering the interactions of entities in a multi-agent dynamical system, where the evaluation is based on the accuracy of future trajectory prediction. Trajectory prediction is in fact an important problem in many multi-agent systems, including the ever growing area of autonomous driving. In fact, our approach falls into the category of *multivariate* time-series prediction (Yu et al., 2018; Wu et al., 2019; Sen et al., 2019; Salinas et al., 2020; Rangapuram et al., 2018; Li et al., 2018; Bai et al., 2018), in which the prediction is based on the *relationship* among the series. Specifically we use the relationship of the entities in the GNNs framework. GNN has been widely used in the trajectory prediction, especially in the application of autonomous driving and significantly improved the performance in this area. For example, Salzmann et al. (2020) uses GNN to capture the relationship among different road users (vehicles and pedestrian), Gao et al. (2020) uses graph attention networks (GATs) to learn the relationship among agents and different components of map data, and Liang et al. (2020) uses graph convolutional networks (GCNs) to learn the interaction among lanes and vehicles.

In our experiment we consider scenarios in which the goal (final) position of the entities is given as the individualized information. In the context of goal-aware prediction, there have been some effort in the area of autonomous driving that are not based on explicit relational learning (Rhinehart et al., 2019; 2018). In these papers, the prediction is based on an autoregressive flow-based model that considers a collective observation of all entities to make prediction for each of them. The goal is then added at the inference time and the latent codes are optimized in a way that the target entity reaches its goal.

## 3 BACKGROUND: NEURAL RELATIONAL INFERENCE (NRI)

NRI is an unsupervised model that learns to infer the interaction types among entities in a multi-agent systems in order to model the dynamics of the system. The model is defined as the problem of predicting the future trajectory of entities given the past trajectories. Formally, the trajectories of $N$ entities are given for $T$ time steps. Entity $i$ is denoted by $\mathbf{x}_i = (\mathbf{x}_i^1, \mathbf{x}_i^2, ..., \mathbf{x}_i^T)$. The set of all trajectories at time $t$ is denoted by $\mathbf{x}^t = \{\mathbf{x}_1^t, \mathbf{x}_2^t, ..., \mathbf{x}_N^t\}$ and $\mathbf{x} = (\mathbf{x}^1, \mathbf{x}^2, ..., \mathbf{x}^T)$ denotes the whole trajectories for all agents. NRI tries to model the system by maximizing the log-likelihood of the observations, $\log p(\mathbf{x})$, in the framework of variational inference, i.e. maximizes the evidence lower-bound (ELBO):

$$\mathcal{L}(\theta, \phi) = \mathbb{E}_{q_\phi(\mathbf{z}|\mathbf{x})}[\log p_\theta(\mathbf{x}|\mathbf{z})] - \text{KL}\big[q_\phi(\mathbf{z}|\mathbf{x})||p(\mathbf{z})\big], \tag{1}$$

where latent variable $\mathbf{z}$ has a categorical distribution and represents the interaction among entities. More specifically, $\mathbf{z}_{ij}$ is a $K$-dimensional vector that denotes the type of interaction between entities $\mathbf{x}_i$ and $\mathbf{x}_j$. The entities in the NRI model are represented using nodes of a graph[1] and therefore the interactions are directed edges on this graph. Parameters of $p(.)$ and $q(.)$ models are denoted by $\theta$ and $\phi$, respectively. The three probability distributions in Eq. 1 are:

- **The variational posterior**, $q_\phi(\mathbf{z}|\mathbf{x})$, is implemented using amortized inference parameterized by a neural network, namely the encoder network. Given the input trajectories, the encoder network predicts the type of edges on the graph. The latent variable $\mathbf{z}$ is assumed to have a categorical distribution. Samples from this distribution form the edges of the graph. In order to backpropagate the error signals to the encoder layers, we need to make the sampling process differentiable. In NRI this is done by approximating the posterior distribution and reparamterezation of Gumbel distribution (Jang et al., 2017; Maddison et al., 2017):

$$\mathbf{z}_{ij} = \text{softmax}((\mathbf{h}_{(i,j)}^2 + \mathbf{g})/\tau) \tag{2}$$

  where $\mathbf{h}_{(i,j)}^2$ is the last output of the encoder before the softmax layer and $\mathbf{g} \in \mathbb{R}^K$ shows i.i.d. samples drawn from Gumbel$(0, 1)$ and $\tau$ is a hyperparameter that controls the smoothness of the distribution.

- **The prior**, $p(\mathbf{z}) = \prod_{i \neq j} p(\mathbf{z}_{ij})$, is assumed to be a factorized uniform categorical distribution over the edges.

- **The likelihood**, $p_\theta(\mathbf{x}|\mathbf{z})$, is implemented by the decoder network and predicts the future trajectories given the uncovered structure of the graph.

The prediction in NRI is done in an autoregressive fashion. However, the ground truth trajectory is fed to the model for few steps during the training to improve the performance of the decoder (teacher forcing). NRI in its original form has two main shortcomings:

1. The latent variable $\mathbf{z}$, which defines the edge types, is fixed for the whole prediction horizon. That is, the uncovered interactions among the agents are assumed to be fixed over the next time steps. This is not necessarily a valid assumption as the agents can dynamically change their interactions in the system.

---

[1]Throughout the paper, entities and nodes as well as interactions and edges are used interchangeably, based on the context.

2. The prior distribution is assumed to be uniform and not conditioned on the previous observations. Both of these assumptions can degrade the performance of the model in longer prediction horizons since samples from the prior provides minimum information about the input.

More recently, Graber & Schwing (2020) pointed out the above issues and addressed them in dynamic neural relation inference (dNRI) model. The conditional prior distribution in dNRI is defined as:

$$p(\mathbf{z}|\mathbf{x}) := \prod_{t=1}^{T} p(\mathbf{z}^t|\mathbf{x}^{1:t}, \mathbf{z}^{1:t-1}), \tag{3}$$

which is implemented by another set of MLP and LSTM layers that form the encoder of the $p_\theta(.)$ model. The experiments show that, by resolving those issues, dNRI achieves better prediction performance than NRI.

## 4 MODEL DESCRIPTION

### 4.1 PROBLEM STATEMENT

We define our problem in the setting of NRI. However, we assume that, in addition to the observable feature set $\mathbf{x}$, each entity $i$ can potentially have access to a set of individualized features $\mathbf{c}_i^t$ at each time step $t$. $\mathbf{c}_i^t$ cannot be observed by other entities [2]. Similar to the observable features, we denote the set of individualized features for each agent by $\mathbf{c}_i = (\mathbf{c}_i^1, \mathbf{c}_i^2, ..., \mathbf{c}_i^T)$, set of individualized features for all agents at time $t$ by $\mathbf{c}^t = \{\mathbf{c}_1^t, \mathbf{c}_2^t, ..., \mathbf{c}_N^t\}$, and set of all individualized features by $\mathbf{c} = (\mathbf{c}^1, \mathbf{c}^2, ..., \mathbf{c}^T)$. We still study the problem of modelling the dynamics of the system through predicting the future trajectories of the entities. However, we aim to exploit the individualized features in a way that the interaction among the entities are inferred more accurately and, therefore, provide a better model for the underlying dynamics of the system, which can lead to better prediction performance.

### 4.2 REPRESENTATION OF INDIVIDUALIZED FEATURES IN THE GRAPH: PRIVATE VS PUBLIC NODES

In order to build the interaction inference model, we need to first represent the individualized features in the graph. Note that simply augmenting the observation features $\mathbf{x}_i$ with the individualized features and building a new set of features, e.g. $\mathbf{y}_i = f(\mathbf{x}_i, \mathbf{c}_i)$, is not a solution here, as this will affect the interaction uncovering among the agents, which is in contrary with our initial assumption about the accessibility of individualized features.

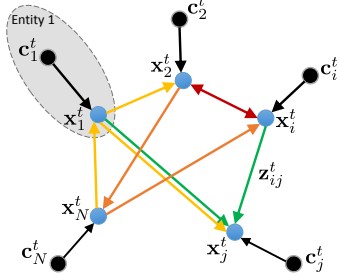

Figure 1: Blue dots and black dots show the public and private nodes, respectively. Different type of interactions among entities are depicted as directed edges among the nodes with different colors.

Here, we propose adding a new set of nodes to represent the individualized features and we call these nodes *private nodes*. We also refer to the nodes that represent the observable nodes as *public nodes* for clarification. Therefore each entity $i$ at each time $t$ can be shown using a public node and private node corresponding to $\mathbf{x}_i^t$ and $\mathbf{c}_i^t$, respectively. A private node is only accessible by its corresponding public node while a public node is accessible by all other public nodes. Interaction among public nodes and their corresponding private node is always fixed, while interaction among public nodes are learned. Fig. 1 shows the uncovered graph of an example system at time step $t$. Note that there is always an edge between the public node and its corresponding private node, which is shown in black in the figure. This edge represents *all* types of interactions. Representing the individualized features using the private nodes allows us to employ a unified framework to uncover the graph, without creating a computational overhead.

---

[2]Although, in some cases the effect of individualized features *might be* observed in the future times steps through $\mathbf{x}_i^{t+k}$ ($k > 0$). For example the intention of a driver is a hidden feature but its effect can be observed through the future trajectory of the car. Note that not all hidden features have this property. Whether this property holds or not does not affect the performance of our model

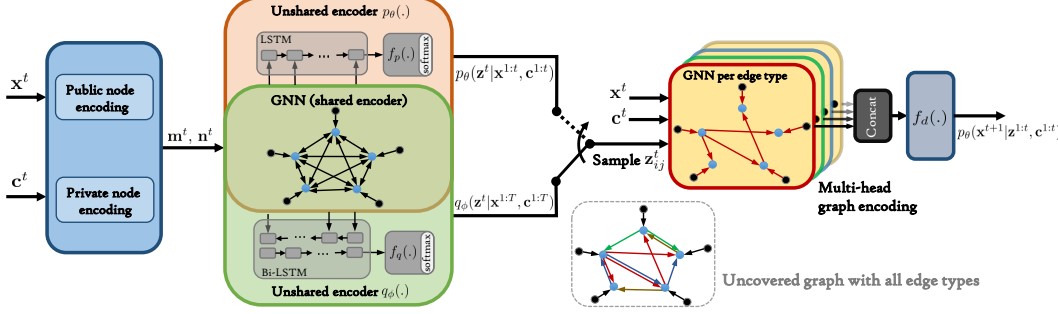

Figure 2: Proposed model. Encoded observable and individualized features are fed to GNN model to update the node features. The output of the GNN is fed to both $p_\theta(.)$ and $q_\phi(.)$ encoders to provide the uncovered interactions among the agents. The parameter of $p_\theta(.)$ and $q_\phi(.)$ distributions are learned to minimize their KL divergence. Moreover, during the training the edges are randomly drawn from the $p_\theta(.)$ encoder to better optimize the parameters of this model. The sampled edges together with the input features are fed to the decoder model to output the prediction at each time step.

## 4.3 MODEL COMPONENTS

In order to learn the interactions in our model we maximize the ELBO of the following form:

$$\mathcal{L}^{\text{NRI-NSI}}(\theta, \phi) = \mathbb{E}_{q_\phi(\mathbf{z}|\mathbf{x},\mathbf{c})}[\log p_\theta(\mathbf{x}|\mathbf{z},\mathbf{c})] - \text{KL}[q_\phi(\mathbf{z}|\mathbf{x},\mathbf{c})||p_\theta(\mathbf{z}|\mathbf{x},\mathbf{c})]. \tag{4}$$

The conditional probability distributions are parameterized by neural networks and factorized as:

$$q_\phi(\mathbf{z}|\mathbf{x},\mathbf{c}) = \prod_{t=1}^{T} q_\phi(\mathbf{z}^t|\mathbf{x}^{1:T}, \mathbf{z}^{1:t-1}, \mathbf{c}^{1:T}) = \prod_{i=1}^{N}\prod_{\substack{j=1\\j\neq i}}^{N}\prod_{t=1}^{T} q_\phi(\mathbf{z}_{ij}^t|\mathbf{x}^{1:T}, \mathbf{z}^{1:t-1}, \mathbf{c}_j^{1:T}), \tag{5}$$

$$p_\theta(\mathbf{z}|\mathbf{x},\mathbf{c}) = \prod_{t=1}^{T} p_\theta(\mathbf{z}^t|\mathbf{x}^{1:t}, \mathbf{z}^{1:t-1}, \mathbf{c}^{1:t}) = \prod_{i=1}^{N}\prod_{\substack{j=1\\j\neq i}}^{N}\prod_{t=1}^{T} p_\theta(\mathbf{z}_{ij}^t|\mathbf{x}^{1:t}, \mathbf{z}^{1:t-1}, \mathbf{c}_j^{1:t}), \tag{6}$$

$$p_\theta(\mathbf{x}|\mathbf{z},\mathbf{c}) = \prod_{t=1}^{T} p_\theta(\mathbf{x}^{t+1}|\mathbf{x}^{1:t}, \mathbf{z}^t, \mathbf{c}^{1:t}) = \prod_{j=1}^{N}\prod_{t=1}^{T} p_\theta(\mathbf{x}_j^{t+1}|\mathbf{x}^{1:t}, \mathbf{z}^t, \mathbf{c}_j^{1:t}). \tag{7}$$

Note that at each time step, the edge uncovering and prediction for each public node depend only on its own private node. Here we describe the details of each component.

### 4.3.1 CONDITIONAL PRIOR NETWORK $p_\theta(\mathbf{z}|\mathbf{x},\mathbf{c})$

The prior distribution forms the different types of interactions given the past and current status of public and private nodes at each time step using the autoregressive model in Eq. 6. We use a GNN architecture for message passing where the graph is fully-connected for the set of public nodes and for the private nodes there is only an edge towards their corresponding public node.

$$\textit{node embedding:} \qquad \mathbf{m}_j^t = f_{\text{emb}}^{\mathbf{x}}(\mathbf{x}_j^t) \;,\;\; \mathbf{n}_j^t = f_{\text{emb}}^{\mathbf{c}}(\mathbf{c}_j^t), \tag{8}$$

$$v \to e: \qquad \mathbf{h}_{(i,j),1}^t = f_e^1([\mathbf{m}_i^t, \mathbf{m}_j^t]), \tag{9}$$

$$e \to v: \qquad \mathbf{h}_j^t = f_v^1(\sum_{i\neq j} \mathbf{h}_{(i,j),1}^t), \tag{10}$$

$$v \to e: \qquad \mathbf{h}_{(i,j),2}^t = f_e^2([\mathbf{h}_i^t, \mathbf{h}_j^t]) \circ f_e^3([\mathbf{h}_i^t, \mathbf{h}_j^t, \mathbf{n}_j^t]), \tag{11}$$

where $\circ$ denotes Hadamard product and $v$ and $e$ represent nodes and edges of the graph, respectively. Note that using our proposed represenetation, we can handle the general setting in which not all entities necessarily have individualized features. In the case that these features are not provided for an entity, $f_e^3(.)$ is masked out and replaced by an all one vector. By using two levels of message

passing we make sure that all observable nodes are considered to infer an edge. In the first level only the observable features are used so that the final relationship uncovering for each entity is not affected by other entities' individualized features. We use multilayer perceptron (MLP) layers for all of the $f(.)$ functions. In order to take into account the previous interactions, the embedding $\mathbf{h}^t_{(i,j),2}$ at each time step is fed to layers of LSTM followed by MLP and softmax to output the actual conditional prior distribution:

$$\mathbf{h}^t_{(i,j),p} = \text{LSTM}(\mathbf{h}^t_{(i,j),2}, \mathbf{h}^{t-1}_{(i,j),p}), \tag{12}$$

$$p_\theta(\mathbf{z}^t_{ij}|\mathbf{x}^{1:t}, \mathbf{z}^{1:t-1}, \mathbf{c}^{1:t}_i) = \text{softmax}\big(f_p(\mathbf{h}^t_{(i,j),p})\big). \tag{13}$$

### 4.3.2 VARIATIONAL POSTERIOR NETWORK $q_\phi(\mathbf{z}|\mathbf{x}, \mathbf{c})$

The variational posterior approximates the true posterior. During the training the graphs are constructed based on the samples from this distribution. According to Eq. 5, we use the whole set of $\mathbf{x}$ and $\mathbf{c}$ from time 1 to $T$ to infer the edges from $q_\phi(\mathbf{z}|\mathbf{x}, \mathbf{c})$. This will result in predicting more accurate edge types during training, compared to the prior model $p_\theta(\mathbf{z}|\mathbf{x}, \mathbf{c})$. Minimizing the KL divergence in Eq. 4 allows us to transfer knowledge about the future to the prior model by minimzing the gap between $p_\theta(\mathbf{z}|\mathbf{x}, \mathbf{c})$ and $q_\phi(\mathbf{z}|\mathbf{x}, \mathbf{c})$. We employ two ideas to help this minimization:

- Parameter sharing between $q_\phi(\mathbf{z}|\mathbf{x}, \mathbf{c})$ and $p_\theta(\mathbf{z}|\mathbf{x}, \mathbf{c})$: We use the same networks in Eq. 8-11 to obtain the edge information. Then a bi-directional LSTM is used to combine this information from previous and future time steps. The output goes to an MLP layer followed by softmax. This will allow identical representation learning for the shared part and the KL divergence minimization should be only applied to the unshared parts of the two models. Note that this is essentially different from the model in Graber & Schwing (2020), in which $p_\theta(\mathbf{z}|\mathbf{x}, \mathbf{c})$ is basically a sub-module of $q_\phi(\mathbf{z}|\mathbf{x}, \mathbf{c})$ and therefore variational posterior cannot contribute in learning a better edge uncovering beyond the prior model. Our variational distribution is given by:

$$\mathbf{h}^t_{(i,j),q} = \text{Bi-LSTM}(\mathbf{h}^t_{(i,j),2}, \mathbf{h}^{t-1}_{(i,j),q}, \mathbf{h}^{t+1}_{(i,j),q}) \tag{14}$$

$$q_\phi(\mathbf{z}^t_{ij}|\mathbf{x}^{1:t}, \mathbf{z}^{1:t-1}, \mathbf{c}^{1:t}_i) = \text{softmax}\big(f_q(\mathbf{h}^t_{(i,j),q})\big). \tag{15}$$

We use reparameterization trick with Gumbel distribution to backpropagate error during the training.

- Sampling from the prior during the training: In order to further close the gap between the two distributions, $\alpha\%$ of the time the edges are sampled from the prior distribution and fed to the decoder. Therefore the parameters of the prior encoder are trained with the prediction loss as well as the KL divergence loss.

### 4.3.3 DECODER NETWORK $p_\theta(\mathbf{x}|\mathbf{z}, \mathbf{c})$

The decoder network, takes the uncovered graph as the input and makes predictions by running a GNN-based algorithm on different edge types. The outputs are then concatenated and go through a decoder function, $f_d(.)$, which is an LSTM for autoregresive prediction. Note that at each time step the private nodes affect both edge uncovering in the encoder and updating nodes states for prediction in the decoder. Similar to NRI, in order to improve the optimization of the decoder parameters during the training, we use teacher forcing and feed the model with ground truth (instead of previous predictions) for the first 10 time steps.

$$v \to e: \qquad \hat{\mathbf{h}}^t_{(i,j),k} = z_{ij,k}\big(\hat{f}^k_1([\mathbf{x}^t_i, \mathbf{x}^t_j]) \circ \hat{f}^k_2([\mathbf{x}^t_i, \mathbf{x}^t_j, \mathbf{c}^t_j])\big), \tag{16}$$

$$e \to v: \qquad \hat{\mathbf{h}}^t_{j,k} = \hat{f}^k_v(\sum_{i \neq j} \hat{\mathbf{h}}^t_{(i,j),k}), \tag{17}$$

$$\boldsymbol{\mu}^{t+1}_j = \mathbf{x}^t_j + \text{LSTM}(\text{Concat}_k[\hat{\mathbf{h}}^t_{j,k}], \boldsymbol{\mu}^t_j), \tag{18}$$

$$p_\theta(\mathbf{x}^{t+1}_j|\mathbf{x}^{1:t}, \mathbf{z}^t, \mathbf{c}^{1:t}_j) = \mathcal{N}(\boldsymbol{\mu}^{t+1}_j, \sigma^2\mathbf{I}). \tag{19}$$

where $k$ is the type of interaction, $\sigma$ is a fixed variance and is set during the training using the validation set, and all $\hat{f}(.)$ functions are implemented by layers of MLP.

## 5 EXPERIMENTS

We perform our experiments on two different tasks, i.e. goal-conditional prediction and action-conditional prediction. For both tasks we consider a multi-agent system in which at least one agent has access to its individualized features. Both of these tasks are of great interest in the context of trajectory prediction, with important downstream applications such as planning. For the goal-conditional task the individualized feature is the final goal (position) of the agent. Therefore, this information is fixed for the whole prediction horizon or at least for multiple time steps, $c_i^{t:t+l} = g_i^t$ for $l > 1$. For the action-conditional task the individualized feature is the next action of the agent, which changes at every time step, $c_i^t = u_i^t$. We refer to our model as NRI-NSI. In all of our experiment we use ADAM optimizer (Kingma & Ba, 2015) with learning rate 0.0001.

**Metrics:** Since our final task is trajectory prediction of the entities, we use minimum average displacement error (minADE) and minimum final displacement error (FDE) as the evaluation metrics. **ADE:** average mean square error (MSE) over all time steps between the ground truth future trajectory and the predicted trajectory. **FDE**: MSE between the final ground truth position and the predicted final position. Note that since we are using a stochastic model in the decoder, the minADE and minFDE are the closest sample to the ground truth over 20 different sampled predictions. We follow this scheme for all of the baselines, too.

**Baselines:** For both action-conditional and goal-conditional prediction, we compare our model with NRI and dNRI where the individualized features are fed to the model at the last stage of decoder for each of the entities, i.e. before outputting the distribution of the predictions.

### 5.1 ACTION-CONDITIONAL PREDICTION

For this task we further consider FM-MPUR (Henaff et al., 2019) as a baseline. FM-MPUR is also an action-conditional prediction model based on conditional variational autoencoders (CVAEs) that aims to maximize the lower bound of $\log p(\mathbf{x}^{t+1:T}|\mathbf{x}^{1:t}, \mathbf{u}_{ego}^{t:T-1})$. The model uses high-dimensional features (images) of the environment that help reasoning about the interaction of the agents.

#### 5.1.1 NGSIM I-80 DATASET:

We study this problem in the domain of autonomous driving. The Next Generation Simulation program's Interstate 80 (NGSIM I-80) (Halkias & Colyar, 2006) dataset consists of 3 batches of 15-minute recordings from traffic cameras mounted over a stretch of a highway in the US. Driving behaviours are complex with complicated interactions between vehicles moving at high-speed. We consider 7 different agents at each time step, where the state of each agent is given by its coordinates and velocity. We assume that for one of the agents (ego-agent), the action is given for every time steps. Actions are two-dimensional vectors that represent acceleration and rotation of the steering wheel. The other 6 vehicles are the closest ones to the ego-agent in different lanes (same lane/left lane/right lane) at time $t = 1$. We keep the state of these vehicles until the end of prediction horizon. We consider 4 relation types for NRI, dNRI, and NRI-NSI.

Table 1 shows the results of different models. We sample the data at 2Hz and perform the prediction task for different prediction horizons. FM-MPUR performs worse than dNRI and NRI-NSI, due to the lack of explicit dynamic interaction modelling. To better illustrate the capability of our model, we consider two different scenarios for NRI, dNRI, and NRI-NSI. First, we use the models without the actions and then add the actions to see how this affect the results and how efficient is our modelling of the individualized features. dNRI outperform NRI for both scenarios, but both of models almost fail to take advantage of the additional information about the actions. On the other hand, NRI-NSI outperform all of the baselines for both with and without action scenarios and obtain a significant gain by adding the actions, meaning that our model can predict more accurate interactions given the actions of the ego-agent. Also our model outperforms dNRI even in the scenario with no actions, which can be associated to the better learning of the prior encoder, as described in section 4.3.2. Moreover, to see how this additional information changes the prediction of the agents with and without the individualized features, we report the results for the set of *all* agents, which includes the ego-agent, and the set of *other* agents which excludes the ego-agent. Given the action of the ego-agent, approximating the next state of this agent is easy. Therefore we can see that results for all

| Model | minADE/minFDE (all - other) | | | |
|---|---|---|---|---|
| | 1s | 2s | 3s | 4s |
| FM-MPUR | 0.34/0.84 - 0.38/0.94 | 0.41/1.06 - 0.45/1.19 | 0.54/1.43 - 0.61/1.68 | 0.64/1.75 - 0.70/1.85 |
| NRI no action | 0.39/0.95 | 0.50/1.17 | 0.68/1.78 | 0.75/2.01 |
| NRI with action | 0.36/0.88 - 0.38/0.91 | 0.44/1.05 - 0.49/1.19 | 0.62/1.67 - 0.67/1.83 | 0.70/1.93 - 0.75/2.02 |
| dNRI no action | 0.33/0.85 | 0.39/1.03 | 0.49/1.24 | 0.57/1.65 |
| dNRI with action | 0.27/0.66 - 0.32/0.72 | 0.34/0.95 - 0.38/1.03 | 0.43/1.18 - 0.48/1.30 | 0.51/1.34 - 0.57/1.54 |
| NRI-NSI no action | 0.28/0.61 | 0.33/0.90 | 0.42/1.03 | 0.49/1.52 |
| NRI-NSI with action | **0.20/0.51 - 0.23/0.58** | **0.26/0.63 - 0.29/0.69** | **0.34/0.75 - 0.37/0.83** | **0.42/1.02 - 0.45/1.17** |

Table 1: Performance of models on action-conditional task for different prediction horizons

| Model | minADE/minFDE (all - other) | | | |
|---|---|---|---|---|
| | 1s | 2s | 3s | 4s |
| NRI | 0.11/0.13 - 0.12/0.16 | 0.14/0.20 - 0.16/0.24 | 0.24/0.38 - 0.28/0.43 | 0.30/0.60 - 0.35/0.68 |
| dNRI | 0.09/0.12 - 0.10/0.14 | 0.12/0.17 - 0.15/0.23 | 0.20/0.31 - 0.25/0.35 | 0.27/0.50 - 0.34/0.58 |
| Trajectron++ | 0.10/0.13 - 0.10/0.15 | 0.12/0.18 - 0.14/0.22 | 0.19/0.33 - 0.23/0.39 | 0.28/0.54 - 0.34/0.60 |
| PRECOG | **0.06/0.08 - 0.06/0.09** | 0.09/0.15 - 0.10/0.18 | 0.16/0.25 - 0.18/0.29 | 0.25/0.48 - 0.29/0.58 |
| NRI-NSI | **0.06/0.08 - 0.06/0.09** | **0.07/0.11 - 0.08/0.12** | **0.12/0.19 - 0.13/0.21** | **0.18/0.31 - 0.20/0.36** |

Table 2: Performance of models on goal-conditional task for different prediction horizons on the Basketball dataset.

| Model | minADE/minFDE (all-other) | | | |
|---|---|---|---|---|
| | 1s | 2s | 3s | 4s |
| NRI | 0.27/0.54 - 0.31/0.60 | 0.38/0.80 - 0.42/0.88 | 0.55/1.24 - 0.63/1.40 | 0.88/1.98 - 0.98/2.12 |
| dNRI | 0.23/0.52 - 0.27/0.54 | 0.34/0.77 - 0.42/0.84 | 0.51/1.08 - 0.59/1.22 | 0.78/1.62 - 0.86/1.80 |
| Trajectron++ | 0.24/0.49 - 0.27/0.52 | 0.35/0.74 - 0.39/0.77 | 0.48/0.97 - 0.54/1.08 | 0.76/1.68 - 0.83/1.97 |
| PRECOG | 0.21/0.47 - 0.24/0.49 | 0.33/0.68 - 0.37/0.74 | 0.44/0.85 - 0.49/0.96 | 0.73/1.55 - 0.82/1.73 |
| NRI-NSI | **0.18/0.37 - 0.18/0.38** | **0.27/0.54 - 0.29/0.56** | **0.38/0.78 - 0.40/0.80** | **0.62/1.34 - 0.64/1.37** |

Table 3: Performance of models on goal-conditional task for different prediction horizons on the nuScenes dataset.

models gets worse by removing the ego-agent. Note that for the scenario without the actions these two results are the same.

## 5.2 GOAL-CONDITIONAL PREDICTION

We consider the goal-conditional prediction on two different sets of data with different types of interactions, i.e. a basketball dataset and an autonomous driving dataset. For this task we also consider two other baselines. Trajectron++ (Salzmann et al., 2020), which uses spatio-temporal information for relational reasoning. It employs dynamical models for the agents in the scene to produce feasible trajectories for the entities. For this model we also inject the goal information at the last level of decoder for each agent. We also consider PRECOG (Rhinehart et al., 2019), which performs goal-conditional prediction using a flow-based generative model. The goal-conditioned prediction is done by optimizing the samples of the latent code of a trained model.

### 5.2.1 BASKETBALL DATASET

We study a basketball players trajectory dataset (Yue et al., 2014). The dataset consists of trajectories of 5 players in the offensive team. We use the same preprocessing as Graber & Schwing (2020). The trajectories at each time step consist of position and velocity of the players. The length of each trajectory is almost 8 seconds, which is sampled at 5Hz (almost 40 frames per trajectory). All models are trained given 20 frames as the input. The prediction horizon varies from 5 to 20 frames. For this experiment we randomly choose 2 players and assign them their goal as their individualized features. The game is very agile, therefore we update the goal every 3 time steps. For this dataset none of the models use any context information and only use the trajectory of the players. We consider 2 relation types for NRI, dNRI, and NRI-NSI.

Table 2 shows the results for this dataset. Again, we report the results for both all agents and other agents (set of agents with no individualized features). For Trajectron++ , their dynamical model cannot fully capture the motion of the players, therefore we removed this module from the model for this dataset and directly predict the next state. We report the results for different prediction horizons and as we can see NRI-NSI outperforms the baselines significantly. PRECOG also performs well on this task as it also systematically optimize the latent codes for goal-conditional prediction during inference. However, our model outperforms PRECOG almost across all horizons because in NRI-NSI the goal-conditional prediction is done systematically during both training and inference time and also the interaction learning is done explicitly at each time step. Therefore, the performance gap between our model and other baselines enlarges as the prediction horizon increases.

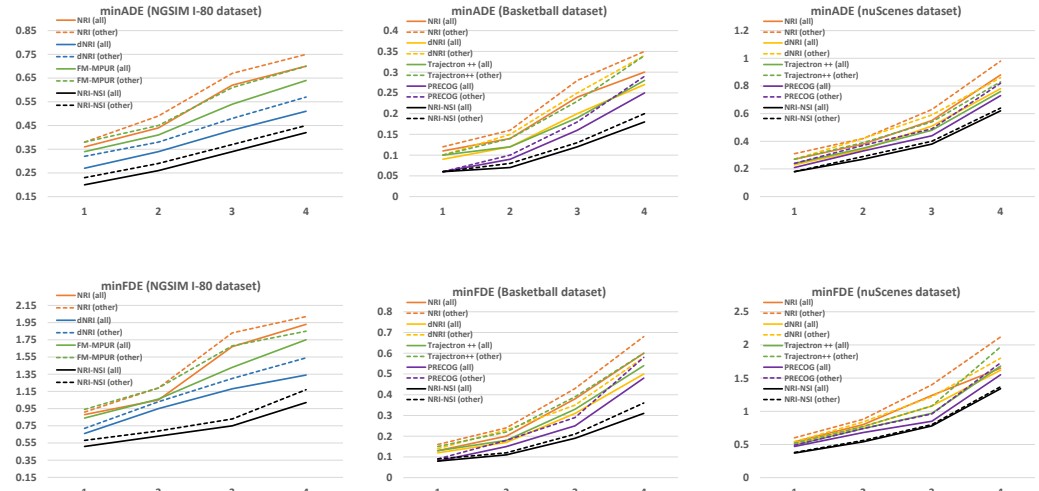

Figure 3: Result of action conditional (left column) and goal-conditional (middle and right columns) experiments on different datasets based on minADE and minFDE for different time horizons. We can see that our model outperforms other model across all tasks. Specially for the goal-conditional tasks the performance of our model drops less than other models from the set of *all* agent to the *other* agents

### 5.2.2 NUSCENES DATASET

nuScenes dataset Caesar et al. (2020), is a real-world dataset for multi-agent trajectory prediction. It consists of 850 episodes of 20 seconds of driving, recorded at 2Hz. The dataset includes the positions of all agents together with synchronized context information, e.g. map. We encode the context information using a convolutional neural network (CNN) and use the encoded information as node in our graph that all entities have access to, similar to Li et al. (2020). This is also done for NRI and dNRI. We use 2 seconds of past trajectories and predict up to 4 seconds into the future. Individualized features are assigned to 2 agents in each scene at random. We set the goal as the last position of the agents for each horizon. Similar to the NGSIM I-80 dataset we use 4 relation types here.

Table 3 shows the results for this model. Similar to the Basketball dataset our model and PRECOG outperform other baselines. Another interesting observation is that our model performs very similarly on the set of *all* and *other* agents. This means that the additional information provided by the goal is effectively propagated to the whole set of agents at each time step (through the *observable features* from the previous steps) and therefore the interactions for all agents are inferred more accurately. However, performance of other baselines is heavily biased in favor of the agents with the individualized features. Therefore, when we remove these agents the values of minADE/minFDE for *other* agents drop more. This can be seen across all tasks in Tables 1-3. Fig. 3 better demonstrates these results.

## 6 CONCLUSION

We considered a novel problem in the framework of relational inference for multi-agent systems in which each entity can potentially have access to a set of individualized information that affects the relationships among the entities. We tackled this problem by assuming a graph structure for the system and proposed to represent the individualized information using private nodes on this graph, whereas the observable features of the agents are the public nodes. We then used variational inference to uncover the edges among the public nodes of the graph. Through experiments with real-world datasets, we showed that our model is capable of making predictions with high accuracy where the individualized information is the next action or the goal of the agents. Note that for all of the baselines the goal of all entities (last position of the entity) is given during the training of the model. However, we showed here that our model can efficiently and explicitly exploit such information to form better interaction among the entities, which is reflected in better prediction accuracy.

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

## A  APPENDIX 1: IMPLEMENTATION DETAILS

### A.1  ARCHITECTURE DETAILS:

Here we provide the architecture of the networks according to Eq. 8-19.

- $f_{\text{emb}}^{\mathbf{x}}$ : 2 layers of MLP with 256 and 128 units and ELU activation
- $f_{\text{emb}}^{\mathbf{c}}$ : 2 layers of MLP with 256 and 128 units and ELU activation
- $f_e^1$ : 2 layers of MLP with 256 and 128 units and ELU activation
- $f_e^2$ : 2 layers of MLP with 256 and 64 units and ELU activation
- $f_e^3$ : 2 layers of MLP with 256 and 64 units and ELU activation
- $f_v^1$ : 2 layers of MLP with 256 and 256 units and ELU activation
- LSTM of $p_\theta(\mathbf{z}|\mathbf{x}, \mathbf{c})$: hidden size 64
- $f_p$ : 3 layers of MLP with 256, 256, and (number of edge types) units and ELU activation
- Bi-LSTM of $q_\phi(\mathbf{z}|\mathbf{x}, \mathbf{c})$: hidden size 64
- $f_q$ : 3 layers of MLP with 256, 256, and (number of edge types) units and ELU activation
- $\hat{f}_1^k$ : 2 layers of MLP with 256 and 128 units and ELU activation
- $\hat{f}_2^k$ : 2 layers of MLP with 256 and 128 units and ELU activation
- $\hat{f}_v^k$ : 2 layers of MLP with 256 and 256 units and ELU activation
- LSTM of $p_\theta(\mathbf{x}|\mathbf{z}, \mathbf{c})$: hidden size 64
- In the case we encode the context information: A CNN with 3 layers of is used with kernel size 5 followed by flattening and 2 MLP layers with 256 and 128 units.
- We use batch normalization after the layers of our neural networks.

### A.2  TRAINING DETAILS:

- Data split:
  - For the NGSIM I-80 the training and test data are split according to the preprocessing of Henaff et al. (2019). The training data is divided to 80% training and 20% validation.
  - For the Basketball dataset the data is divided tof 65% training, 10% validation, and 25% test set.
  - For the nuScence dataset the training set is divided to 80% training and 20% validation. The provided validation set is used for test.
- Hyperparameters (chosen by validation):
  - Parameter of the Gumbel distribution $\tau = 0.5$
  - Percentage of samples from the $p_\theta(\mathbf{z}|\mathbf{x}, \mathbf{c})$ model: $\alpha = 10\%$
- batch size: 128
- Number of epochs:
  - 50 for the NGSIM I-80 dataset
  - 500 for the Basketball dataset
  - 50 for the nuScences dataset
- Type of GPU: single TITAN X GPU.

## B  APPENDIX 2: MORE RESULTS

### B.1  ERROR BOUNDS ON THE RESULTS:

Tables below show the results of the experiment with standard deviation. The results are based on running the models for 3 times. We split the set of all and other agents into two tables.

| Model | minADE/minFDE (all) | | | |
|---|---|---|---|---|
| | 1s | 2s | 3s | 4s |
| FM-MPUR | 0.34±0.012/0.84±0.024 | 0.41±0.015/1.06±0.029 | 0.54±0.018/1.43±0.034 | 0.64±0.019/1.75±0.050 |
| NRI no action | 0.39±0.012/0.95±0.009 | 0.50±0.018/1.17±0.025 | 0.68±0.023/1.78±0.035 | 0.75±0.028/2.01±0.054 |
| NRI with action | 0.36±0.012/0.88±0.018 | 0.44±0.018/1.0±0.019 | 0.62±0.024/1.67±0.036 | 0.70±0.020/1.93±0.045 |
| dNRI no action | 0.33±0.015/0.85±0.029 | 0.39±0.015/1.03±0.023 | 0.49±0.016/1.24±0.039 | 0.57±0.025/1.65±0.041 |
| dNRI with action | 0.27±0.009/0.66±0.012 | 0.34±0.020/0.95±0.031 | 0.43±0.014/1.18±0.035 | 0.51±0.026/1.34±0.037 |
| NRI-NSI no action | 0.28±0.010/0.61±0.011 | 0.33±0.015/0.90±0.019 | 0.42±0.019/1.03±0.025 | 0.49±0.025/1.52±0.046 |
| NRI-NSI with action | **0.20±0.010/0.51±0.022** | **0.26±0.013/0.63±0.020** | **0.34±0.011/0.75±0.012** | **0.42±0.015/1.02±0.033** |

Table 4: Performance of models on action-conditional task for different prediction horizons on the NGSIM I-80 dataset.

| Model | minADE/minFDE (other) | | | |
|---|---|---|---|---|
| | 1s | 2s | 3s | 4s |
| FM-MPUR | 0.38±0.011/0.94±0.012 | 0.45±0.023/1.19±0.030 | 0.61±0.019/1.68±0.036 | 0.70±0.028/1.85±0.044 |
| NRI with action | 0.38±0.012/0.91±0.030 | 0.49±0.015/1.19±0.035 | 0.67±0.019/1.83±0.055 | 0.75±0.023/2.02±0.061 |
| dNRI with action | 0.32±0.010/0.72±0.022 | 0.38±0.014/1.03±0.039 | 0.48±0.015/1.30±0.044 | 0.57±0.024/1.54±0.046 |
| NRI-NSI with action | **0.23±0.015/0.58±0.013** | **0.29±0.015/0.69±0.018** | **0.37±0.015/0.83±0.025** | **0.45±0.013/1.17±0.031** |

Table 5: Performance of models on action-conditional task for different prediction horizons on the NGSIM I-80 dataset.

| Model | minADE/minFDE (all) | | | |
|---|---|---|---|---|
| | 1s | 2s | 3s | 4s |
| NRI | 0.11±0.008/0.13±0.009 | 0.14±0.008/0.20±0.011 | 0.24±0.012/0.38±0.015 | 0.30±0.012/0.60±0.025 |
| dNRI | 0.09±0.005/0.12±0.007 | 0.12±0.011/0.17±0.010 | 0.20±0.011/0.31±0.012 | 0.27±0.013/0.50±0.016 |
| Trajectron++ | 0.10±0.004/0.13±0.008 | 0.12±0.008/0.18±0.013 | 0.19±0.018/0.13±0.013 | 0.28±0.015/0.54±0.019 |
| PRECOG | **0.06±0.006/0.08±0.004** | 0.09±0.005/0.15±0.010 | 0.16±0.012/0.15±0.011 | 0.25±0.013/0.48±0.010 |
| NRI-NSI | **0.06±0.007/0.08±0.005** | **0.07±0.009/0.11±0.009** | **0.12±0.010/0.19±0.014** | **0.18±0.017/0.31±0.013** |

Table 6: Performance of models on goal-conditional task for different prediction horizons on the Basketball dataset.

| Model | minADE/minFDE (other) | | | |
|---|---|---|---|---|
| | 1s | 2s | 3s | 4s |
| NRI | 0.12±0.007/0.16±0.007 | 0.16±0.009/0.24±0.011 | 0.28±0.015/0.43±0.010 | 0.35±0.025/0.68±0.036 |
| dNRI | 0.10±0.006/0.14±0.011 | 0.15±0.007/0.23±0.012 | 0.25±0.016/0.35±0.011 | 0.34±0.020/0.58±0.038 |
| Trajectron++ | 0.10±0.004/0.15±0.009 | 0.14±0.010/0.22±0.014 | 0.23±0.015/0.39±0.010 | 0.34±0.011/0.60±0.015 |
| PRECOG | **0.06±0.004/0.09±0.004** | 0.10±0.006/0.18±0.009 | 0.18±0.011/0.29±0.011 | 0.29±0.015/0.58±0.015 |
| NRI-NSI | **0.06±0.005/0.09±0.004** | **0.08±0.007/0.12±0.006** | **0.13±0.009/0.21±0.015** | **0.20±0.014/0.36±0.013** |

Table 7: Performance of models on goal-conditional task for different prediction horizons on the Basketball dataset.

| Model | minADE/minFDE (all) | | | |
|---|---|---|---|---|
| | 1s | 2s | 3s | 4s |
| NRI | 0.27±0.012/0.54±0.010 | 0.38±0.012/0.80±0.023 | 0.55±0.012/1.24±0.021 | 0.88±0.022/1.98±0.032 |
| dNRI | 0.23±0.012/0.52±0.013 | 0.34±0.011/0.77±0.022 | 0.51±0.011/1.08±0.023 | 0.78±0.018/1.62±0.015 |
| Trajectron++ | 0.24±0.018/0.49±0.014 | 0.35±0.015/0.74±0.011 | 0.48±0.012/0.97±0.025 | 0.76±0.038/1.68±0.051 |
| PRECOG | 0.21±0.010/0.47±0.012 | 0.33±0.015/0.68±0.012 | 0.44±0.019/0.85±0.017 | 0.73±0.029/1.55±0.043 |
| NRI-NSI | **0.18±0.015/0.37±0.012** | **0.27±0.015/0.54±0.016** | **0.38±0.016/0.78±0.017** | **0.62±0.014/1.34±0.028** |

Table 8: Performance of models on goal-conditional task for different prediction horizons on the nuScenes dataset.

| Model | minADE/minFDE (other) | | | |
|---|---|---|---|---|
| | 1s | 2s | 3s | 4s |
| NRI | 0.31±0.011/0.60±0.010 | 0.42±0.010/0.88±0.015 | 0.63±0.015/1.40±0.018 | 0.98±0.018/2.12±0.012 |
| dNRI | 0.27±0.015/0.54±0.016 | 0.42±0.017/0.84±0.022 | 0.59±0.013/1.22±0.019 | 0.86±0.026/1.80±0.045 |
| Trajectron++ | 0.27±0.016/0.52±0.017 | 0.39±0.010/0.77±0.020 | 0.54±0.019/1.08±0.033 | 0.83±0.026/1.97±0.038 |
| PRECOG | 0.24±0.011/0.49±0.019 | 0.37±0.018/0.74±0.015 | 0.49±0.010/0.96±0.029 | 0.82±0.024/1.73±0.041 |
| NRI-NSI | **0.18±0.010/0.38±0.015** | **0.29±0.016/0.56±0.010** | **0.40±0.012/0.80±0.024** | **0.64±0.011/1.37±0.030** |

Table 9: Performance of models on goal-conditional task for different prediction horizons on the nuScenes dataset.

## B.2 Results on relational inference

The results in the tables 1-3 show that NRI-NSI performs better than other baselines in terms of trajectory prediction performance. This might implicitly indicate a better relational inference by NRI-NSI. Nevertheless, here we try to investigate the performance of NRI-NSI compared to NRI and dNRI in terms of relational inference explicitly using a toy dataset.

Here we use the charged particles dataset. We use 5 and 10 particles in a 2D box with positive and negative charges $\{\pm q\}$ that are sampled uniformly. The force among them follow the Coulomb's law. We follow similar procedure by Kipf et al. (2018) to stabilize the generation process, i.e. soft clipping the force using the softplus function. Similarly, we generate 50k training examples, and 10k validation and test examples. Although the generated data might not exactly follow the physics rules, we will have *explicit* relations (force) among the particles and can use this as the ground truth for our experiments. Our goal is to accurately infer the relation among the particles given that we know the final position of 1 of the particles for 5-particle experiment and final position of 2 particles for 10-particle experiment. For all models we use 2 edge types. For both NRI and dNRI the goal information is fed to the last layer of their encoders. Table below shows the accuracy of different models in predicting the relations among the particles:

| Toy dataset | NRI | dNRI | NRI-NSI |
|---|---|---|---|
| 5 particles | $82.3 \pm 0.5$ | $83.1 \pm 0.6$ | $\mathbf{88.5 \pm 0.4}$ |
| 10 particles | $70.6 \pm 0.5$ | $70.9 \pm 0.8$ | $\mathbf{76.8 \pm 0.6}$ |

Table 10: Performance in terms of relational inference accuracy (in %) for the charged particle dataset.

Note that for this results we only used the encoder of different models.

