# OpenReview forum: "Neural Relational Inference with Node-Specific Information "
_ICLR.cc/2022/Conference — ICLR 2022 Poster_

### Official Review · Reviewer_NM6N · 2021-10-20

**Correctness:** 3
**Technical Novelty And Significance:** 3
**Empirical Novelty And Significance:** 3
**Recommendation:** 8
**Confidence:** 3

**Main Review:**

This paper presents a neural relational inference model with node-specific information. Experiments on real-world datasets validate the merit of the proposed method.

Stengths:
1. This paper is well written and the idea is clear.
2. The results are promising compared to various baslines.

Weaknesses:
1. The codes are not availabel, it is difficut for others to re-produce the results.

**Summary Of The Paper:**

This paper presents a neural relational inference model with node-specific information. Experiments on real-world datasets validate the merit of the proposed method.

**Summary Of The Review:**

This paper presents a neural relational inference model with node-specific information. Experiments on real-world datasets validate the merit of the proposed method.

Stengths:
1. This paper is well written and the idea is clear.
2. The results are promising compared to various baslines.

Weaknesses:
1. The codes are not availabel, it is difficut for others to re-produce the results.

---

> ### Author Response · Authors · 2021-11-16
> **Response to Reviewer NM6N**
>
> We thank the reviewer for their valuable time and their feedback. We are happy to see their approval but also would like to elaborate on the issue about the code.
>
> The code for the submission will be available as soon as possible but the implementation of the model should be straightforward given the details in Appendix A.

---

### Official Review · Reviewer_hKEW · 2021-11-03

**Correctness:** 4
**Technical Novelty And Significance:** 2
**Empirical Novelty And Significance:** 2
**Recommendation:** 5
**Confidence:** 2

**Main Review:**

## Weaknesses
* The task assumption of the paper does not look straightforward to me. I am doubtful about the practical value of the task setting: "the individualized information cannot be accessed by other entities but its effect will be observed in the future". Is this a well-recognized setting? Are there practical scenarios of this assumption?
* Given the existing work of NRI and the variational inference framework, the paper is not clear about the contributions. For instance, what are the new challenges that need to be solved after introducing the private entities? Besides the model architectures, what are the key innovations?

## Strengths
* The experimental results look strong and thorough. Different datasets are considered and the improvements are significant.

**Summary Of The Paper:**

**Summary**: This paper introduces a neural relational inference model that makes use of the hidden features of each node in a variational inference framework. Specifically, the hidden/individual information is modeled as private node in the graph. Importantly, the task assumption made by the authors is that these individualized features cannot be observed by other entities.

**Contributions**:
1. The authors claim be to the first to study the use of individualized information for each entity in this direction.
2. The proposed approach achieve state-of-the-art results while only introducing minimum additional computational complexity.

**Summary Of The Review:**

Overall, this paper has demonstrated strong results on neural relational inference tasks. The concerns I have are about the assumption of the approach and unclear novelty points.

---

> ### Author Response · Authors · 2021-11-16
> **Response to Reviewer hKEW**
>
> We thank the reviewer for their valuable time and feedback. We hope to be able to address the raised concerns here and improve your final rating.
>
> Weaknesses:
>
> * The exact quote from the paper reads “ $\mathbf{c}_i^t$ cannot be observed by other entities, however its effect $\textit{might be}$ observed in the future time steps through $\mathbf{x}^{t+k}_i (k > 0)$.” To better present this setting, consider a car whose intention is to change lanes in a highway. This intention is never known to the other cars. However, the intention is reflected in the future trajectory of the car in the next time steps and the other cars can also observe this trajectory. In this example the intention is NSI ($\mathbf{c}_i^t$) and the future trajectory is the effect of this NSI and represented as $\mathbf{x}^{t+k}_i $. This is in fact a practical setting in real-world.
> Please note that there are other forms of NSI that other agents cannot observe their effect, e.g. POV sensory data.
>
>     Our goal in this paper is to best exploit the NSI to form better interactions among the agents so that we can succeed in the downstream tasks. Whether the NSI’s effect can be observed in the future or not, does not affect our proposed model or its performance.
>
> * There are multiple novel points in our work compared to NRI:
>
>    * First of all, the introduction of NSI in this framework is novel. NSI can represent various types of private information that affect the interaction modelling. Relational inference is a crucial problem in the domain of multi-agent systems and the introduction of NSI opens up a new set of challenges and opportunities in this direction. We investigated the effect of two important types of NSI that are of great interest in the autonomous driving community, i.e. actions and goals of agents.
>
>    * Secondly, we presented a novel way to represent the NSI by introducing the private nodes in the graph structure. This also includes defining new types of edges among the nodes.
>
>    * The other important and novel point in our model is in designing the edge uncovering process (encoder) and the node update process (decoder) in the presence of private information. We designed our model in a way that the private information is never leaked from one node to other ones.

---

> ### Author Response · Authors · 2021-12-07
> **Response to Reviewer hKEW**
>
> We would really appreciate it if the reviewer could let us know whether their concerns regarding the paper have been resolved or not. We would be happy to continue this discussion if there is any further questions.
>
> Thank you again for your valuable time and feedbacks.

---

### Official Review · Reviewer_SFbP · 2021-11-03

**Correctness:** 3
**Technical Novelty And Significance:** 4
**Empirical Novelty And Significance:** 3
**Recommendation:** 8
**Confidence:** 2

**Main Review:**

Strengths:
- The concept of node-specific information (NSI) seems to be novel and is interesting.
- The paper is well-written. All components are explained in detail.
- The paper demonstrate the effectiveness of the idea. The proposed approach performs better than reference systems that use the private information only at the decoder. Hence, I agree with the conclusion that the approach helps to form better interaction graphs.
- The distinction to prior works, especially NRI, is clear. It is easy for the reader to understand what the contributions of the paper are.

Weaknesses:
- The paper argues that in many real-world examples, "there is a set of hidden features that affect the way entities interact with each other." However, I am not sure how reasonable this is. For instance, in autonomous driving, it seems reasonable that considering the intention of individual entities can help to generate an interaction graph. However, it is not the hidden features that affect the interaction of the entities, but only the public nodes, which are affected by the hidden features.
- The paper uses autonomous driving as an example. However, in autonomous driving, the intention of a car can be shared with other cars via car2car communication. However, sharing this information is limited to system that are capable of this form of communication. Pedestrians, for instance, also have private intentions that cannot be shared easily.
- I am missing an analysis of the learned graphs, since the paper states that their method is better able to learn interaction graphs. Now I am wondering if this difference can be visualized in an example from one of datasets, or even better analyzed in a systematic way.


Questions:
- Question 1: One perhaps crucial part is not yet entirely clear to me: Since the public nodes are connected, and each public node is connected to its private node, isn't it possible that the private information flows to the other nodes in only 2 hops, especially in the decoder? Then, the private information can be shared with other public nodes. The paper states that "... however its [the private node's] effect might be observed in the future times steps ...". But as far as I understand, it is not only the effect of the private information but the private information itself that can be shared. It would be great if the authors or other reviewers could clarify this point.
- Question 2: Furthermore, I am wondering if the approach can actually be used in real-world situations, for instance for autonomous driving. In a real-world setting, each car, and therefore each model, has its own set of hidden information that is not shared with other cars, which means that each model only has access to the private information of a single car. However, the model assumes to have  access to all the private nodes (i.e. it has a global view of the problem, not only a local view), which seems to not fit to the envisioned application setup.

**Summary Of The Paper:**

The paper introduces the concept of node-specific information (NSI) to model that nodes in a graph may have private information that other nodes cannot have access to. The paper uses Neural Relation Inference (NRI), a framework published in 2018 based on variational inference, to uncover the hidden relations of nodes in the graph. For instance, in a driving scenario, different cars can be nodes in a graph with their publicly visible trajectory and their private information about the intention (e.g. desired destination), which is not shared with other nodes. The encoder and decoder in NRI are modified such that NSI stays private and is not shared with other nodes. The paper considers problems that require uncovering the interactions of entities in a multi-agent dynamical system. The evaluation is based on the accuracy of future trajectory prediction. The paper demonstrates the effectiveness of the idea on three different datasets, one action-conditional dataset and two goal-conditional datasets.

**Summary Of The Review:**

To summarize, the idea of introducing node-specific, private information into a graph of multiple interactive systems seems to be novel and interesting and improves the prediction performance. However, it is not entirely clear to me if the private information can be propagated to other nodes (question 1) and if the model can even be applied in the envisioned scenario (question 2). I think the answers to both questions are important to make a reasonable assessment of the paper. Hence, my recommendation is rather tentative and I will update it as soon as the application setup becomes more clear to me.

After rebuttal:
Thanks a lot for the author's replies. I still think the applicability of the idea may be problematic, but I think the idea of NSI is sufficently interesting and novel to be accepted to this conference. Hence, I raise my score to accept.

---

> ### Author Response · Authors · 2021-11-16
> **Response to Reviewer SFbP**
>
> Thank you very much for your valuable time and detailed comments. We hope that we can resolve the raised issues in a constructive discussion.
>
> First of all, thank you for highlighting the positive points about the paper and finding the idea interesting.
>
>
> Weaknesses:
>
> * Regarding the importance of NSI in multi-agent systems: In multi-agent systems the hidden features (NSI) can be seen in different forms. NSI can be as basic and ubiquitous as the POV observations from the environment for each agent. That is, each agent has its own sensory data received from the environment, which differs from other agents and is not shared among agents. In our paper we considered a more complicated form of NSI, i.e. intention. In our experiments we studied the effect of intention on relational inference in two cases: action-conditional and goal-conditional prediction. Both of these problems are important problems in multi-agent systems, especially in autonomous driving.
>
>    Regarding the role of NSI in relational inference: Additionally we believe that both hidden features and observable features heavily affect the type of relations that an agent should make with other agents. For example, consider a scenario in which a car arrives at an intersection. Let’s think of two cases. In the first case the car’s intention is to turn left. It’s important for this car to make relations with the other cars that are coming from the opposite direction. In the second case the car’s intention is to turn right at the intersection. Although the past trajectories (observable features) are the same as case 1, there are other types of relations with another set of cars that should be inferred. Therefore, the intention plays a role in defining the type of relations a car should make with other cars.
>
>
> * This is a very interesting point. As you mentioned, sharing NSI in autonomous driving is a problem, which is far from being solved. In the current state of autonomous driving, it is still very reasonable and realistic to consider intention as some private information. However, given that there is a capability among the agents in the system to share their NSIs, the problem formulation should change. This seems to be an interesting direction for research, which is out of scope of current submission.
>
>
> * In the appendix B.2 of the paper, we provided evidence about the capability of our model in terms of relational inference. Using two experiments on real-world and synthetic data, we showed our model outperforms the base models in terms of accuracy of predicting relations among different agents.
>
>
>
> Questions:
>
> 1- This is an important question. Based on eq. 8-19, at each time step, the NSI is never shared among the agents in the encoder or the decoder: To form the edge $\mathbf{z}^t_{ij}$ ,  only the private information of node $j$ is used. Also, when we predict   $\mathbf{x}^{t+1}_{j}$  from the constructed graph, only $\mathbf{c}_j^{1:t}$ are used and not others' NSI. For the next time step, we form a “new” graph for the system using the updated observations and NSI. From the new graph we predict a new set of observations and so on. Therefore, NSI is never shared among the agents from one time step to the next one. However, the effect of NSI might be reflected in the predicted observations. For example, if at time t, the intention of car $j$ ($\mathbf{c}_j^t$) is to turn left in an intersection, the other cars can see the effect of this intention at future time steps through $\mathbf{x}_j^{t+1:t+k}$. This is a practical assumption in the real world: we do not observe other cars’ intentions but we observe the effect of their intentions in their future trajectories.
>
>
>
>
> 2- Regarding the second question there are two points that we should mention here:
>    * The proposed model supports both presence and absence of NSI for the agents. However, during the training, random assignment of the NSI (e.g. goal) to agents provides more robustness for the model. During the test we can assign the goal to a single car (local view) or multiple cars (global view) and our model can handle both.
>
>
>    * In some applications, e.g. building a driving simulator, we use the global view to predict/control the cars’ movements. Therefore we can assign the NSI to multiple agents.
>
> We really enjoyed reading the comments of the reviewer and hope to be able to answer their questions and improve their final rating.

---

> ### Author Response · Authors · 2021-11-27
> **Thank you for raising your score**
>
> Thank you very much again for your insightful comments about the paper. It is great to see that your initial concerns are mostly addressed and you raised the score.

---

### Decision · Program_Chairs · 2022-01-20

**Decision:**

Accept (Poster)

**Comment:**

This paper extends the Neural Relational Inference framework for probabilistic inference of interaction relations between entities, to a scenario where entities may have private features, which requires modifications of the standard graph encoders and decoders in NRI.

Reviewers appreciated both the model and the overall execution of the paper: the building blocks are clear, the evaluation does its job well. The main doubts are about the applicability of the setting, for which the authors don't provide too many examples. However, the construction is somewhat intuitive, and even in cases where private attributes aren't explicit, it may be valuable to disentangle the shareable attributes this way. We encourage the reviewers to discuss the applicability a bit further.

Typos: (not exhaustive, please doublecheck with a spell checker)
 - multiple occurrences of Gumble instead of Gumbel
 - bottom of pg 4, factorzied -> factorized